# Towards Automation in IVF: Pre-Clinical Validation of a Deep Learning-Based Embryo Grading System during PGT-A Cycles

**DOI:** 10.3390/jcm12051806

**Published:** 2023-02-23

**Authors:** Danilo Cimadomo, Viviana Chiappetta, Federica Innocenti, Gaia Saturno, Marilena Taggi, Anabella Marconetto, Valentina Casciani, Laura Albricci, Roberta Maggiulli, Giovanni Coticchio, Aisling Ahlström, Jørgen Berntsen, Mark Larman, Andrea Borini, Alberto Vaiarelli, Filippo Maria Ubaldi, Laura Rienzi

**Affiliations:** 1Clinica Valle Giulia, GeneraLife IVF, Via De Notaris 2B, 00197 Rome, Italy; 2Department of Biology and Biotechnology “Lazzaro Spallanzani”, University of Pavia, 27100 Pavia, Italy; 3University Institute of Reproductive Medicine, National University of Cordoba, Cordoba 5187, Argentina; 49.baby, GeneraLife IVF, 40125 Bologna, Italy; 5Livio, GeneraLife IVF, 40229 Göteborg, Sweden; 6Vitrolife A/S, 8260 Aarhus, Denmark; 7Vitrolife Sweden AB, 421 32 Göteborg, Sweden; 8Department of Biomolecular Sciences, University of Urbino “Carlo Bo”, 61029 Urbino, Italy

**Keywords:** IVF, PGT-A, artificial intelligence, automation, time-lapse microscopy, embryo selection, blastocyst, embryo assessment, euploidy, implantation

## Abstract

Preimplantation genetic testing for aneuploidies (PGT-A) is arguably the most effective embryo selection strategy. Nevertheless, it requires greater workload, costs, and expertise. Therefore, a quest towards user-friendly, non-invasive strategies is ongoing. Although insufficient to replace PGT-A, embryo morphological evaluation is significantly associated with embryonic competence, but scarcely reproducible. Recently, artificial intelligence-powered analyses have been proposed to objectify and automate image evaluations. iDAScore v1.0 is a deep-learning model based on a 3D convolutional neural network trained on time-lapse videos from implanted and non-implanted blastocysts. It is a decision support system for ranking blastocysts without manual input. This retrospective, pre-clinical, external validation included 3604 blastocysts and 808 euploid transfers from 1232 cycles. All blastocysts were retrospectively assessed through the iDAScore v1.0; therefore, it did not influence embryologists’ decision-making process. iDAScore v1.0 was significantly associated with embryo morphology and competence, although AUCs for euploidy and live-birth prediction were 0.60 and 0.66, respectively, which is rather comparable to embryologists’ performance. Nevertheless, iDAScore v1.0 is objective and reproducible, while embryologists’ evaluations are not. In a retrospective simulation, iDAScore v1.0 would have ranked euploid blastocysts as top quality in 63% of cases with one or more euploid and aneuploid blastocysts, and it would have questioned embryologists’ ranking in 48% of cases with two or more euploid blastocysts and one or more live birth. Therefore, iDAScore v1.0 may objectify embryologists’ evaluations, but randomized controlled trials are required to assess its clinical value.

## 1. Introduction

Embryo assessment and selection continues to be a major challenge in IVF, especially since IVF clinics started more commonly adopting a single embryo transfer (SET) policy [1]. In fact, embryologists worldwide strive to implement effective strategies to improve IVF efficiency (i.e., higher live birth rate (LBR) per transfer with less risks, efforts, and possibly costs) while preserving its efficacy (i.e., the cumulative live birth delivery rate (CLBdR) per cycle) [2]. Static embryo morphological assessment is still the predominant non-invasive embryo selection strategy used. It consists of several static microscopic observations at fixed time points of preimplantation development focused on a few prognostic features [3]. At the blastocyst stage, the Gardner’s score is the most applied grading system. It is a three-part scoring system based on the degree of blastocyst expansion, inner cell mass (ICM), and trophectoderm (TE) morphology [4]. Blastocyst transfer elicits higher LBR per ET than cleavage stage ET with the same CLBdR per cycle and miscarriage rate per clinical pregnancy [5]. In addition, a significant correlation exists between blastocyst quality with euploidy and implantation in the context of both untested and euploid ET [6,7,8]. Regardless, static assessment suffers from several inherent limitations. Firstly, it is undermined by high subjectivity and both intra- and inter-operator variability [9,10,11]. Moreover, a few snapshots of embryo development cannot provide a complete evaluation of this complex and dynamic process and fail to capture abnormal events, such as abnormal fertilization and cleavage patterns, blastomere exclusion/extrusion, or spontaneous blastocyst collapse [12]. In fact, blastocysts classified as excellent/good quality are often aneuploid or fail to implant, just like blastocysts classified as poor quality (less than Gardner’s BB grade) may actually be euploid and implant [13,14,15,16,17]. 

Implementation of preimplantation genetic testing for aneuploidies (PGT-A) allows the discrimination of chromosomally normal (euploid) from abnormal (aneuploid) embryos in a cohort of blastocysts produced during IVF through biopsy and analysis via comprehensive chromosome testing (CCT) technologies (e.g., q-PCR or NGS) of 5–10 cells from the TE [18]. In the hands of expert operators and well-equipped laboratories, TE biopsy does not negatively impact embryo viability [19,20]. Crucially, the transfer of euploid blastocysts in randomized controlled (RCT) or observational trials involves higher LBR per ET and lower miscarriage rate per clinical pregnancy with respect to untested blastocyst transfer [21,22]. In fact, when blastocysts affected by full-chromosome meiotic aneuploidies were transferred in blinded non-selection studies, they resulted in >98% embryo lethality rate with almost 90% of clinical pregnancies ending up in a miscarriage. Still, this technique requires extensive expertise and several euploid blastocysts fail to result in a live birth (LB), despite a predictive value on implantation as high as 65% [20].

More recently, the introduction of time-lapse technology (TLT) in IVF has allowed continuous monitoring of embryos, undisturbed culture, and precise reporting of developmental timings and abnormal cleavage patterns [12,23,24]. Nevertheless, the data about the true effectiveness of TLT for embryo selection purposes are controversial [25,26]. In fact, embryo morphodynamics is associated with, but cannot effectively predict, euploidy [27,28].

The latest development in this scenario is the combination of artificial intelligence (AI) with TLT. AI leverages computers and machines to mimic human problem-solving and decision-making capabilities. The definition of AI includes machine learning (ML) and deep learning (DL). ML is a data processing technology that can make predictions based on previously analyzed, structured, or labeled data. DL works as a set of neural networks, inspired by the human brain, to learn and detect features from large amounts of unlabeled data. The use of algorithms to guide human decisions would contribute greatly to achieving standardization in IVF, and thus, obtaining more consistent, comparable, and reproducible results, by preventing subjectivity in the evaluation process [29,30,31]. To that end, several systems have been developed lately that can assess individual embryos, segmenting and grading important developmental features, and generating a score (e.g., [32,33,34]). In theory, the AI-powered TLT assessment is a goldmine of information potentially useful for embryo selection purposes [23,27,30,35,36,37,38], but its clinical utility must be tested in properly designed studies and/or large real-life datasets. The software “intelligent data analysis (iDA) Score v1.0” (Vitrolife A/S), which can be directly integrated into an EmbryoScope+ incubator (Vitrolife A/S), is one of these tools. This software (a DL algorithm trained on hundreds of thousands of videos from implanted and non-implanted embryos) generates a score for each embryo from 1.0 to 9.9, which should be representative of its chance to implant. 

We designed this study to assess the degree of concordance between iDAScore v1.0 with: (i) blastocyst morphological assessment carried out by senior embryologists according to Gardner’s criteria, and (ii) blastocysts’ chromosomal constitution (euploidy, single aneuploidy, multiple aneuploidies). We also assessed how often iDAScore v1.0 would have ranked as top-quality euploid blastocysts among cohorts characterized by sibling euploid and aneuploid embryos. Lastly, we assessed in a retrospective simulation how often ranking for transfer of multiple euploid blastocysts by iDAScore v1.0 would have involved an earlier or later LB.

## 2. Material and Methods

### 2.1. Study Design

This is a retrospective study aimed at a pre-clinical validation of iDAScore v1.0 software in PGT-A cycles conducted between April 2013 and August 2022 at a private IVF clinic (Clinica Valle Giulia, GeneraLife IVF, Rome, Italy). Overall, we included 1232 PGT-A cycles with ≥1 biopsied blastocyst (N = 3604 embryos) after undisturbed culture in EmbryoScope incubators (Vitrolife A/S, Aarhus, Denmark). All patients were included only once for their first PGT-A cycle conducted with their own fresh oocytes. Patients with an indication for PGT for structural rearrangements (PGT-SR) and PGT for monogenic conditions (PGT-M) were not included. 

All videos were retrospectively assessed through the iDAScore v1.0 software to grade each blastocyst without influencing embryologists’ clinical evaluations and decision-making process. iDAScore v1.0 was then assessed for its concordance with blastocyst quality as defined by senior embryologists according to the Gardner’s criteria (ICM morphology, TE morphology, and overall blastocyst morphology), and day of biopsy defined according to the hpi (≤120 hpi = day 5, 121–144 = day 6, >144 = day 7) (Figure 1). Overall, 771 patients obtained ≥1 euploid blastocyst (N = 1443 embryos) after TE biopsy-based chromosomal testing conducted via CCT technologies at an external genetic laboratory (Igenomix, Marostica, Italy). iDAScore v1.0 was also assessed for its association with blastocyst karyotype (Figure 1) categorized as euploid, single aneuploid (i.e., single monosomy or single trisomy), or complex aneuploid (i.e., ≥2 aneuploidies). A sub-analysis of iDAScore v1.0 in these three groups was conducted within blastocyst quality categories as defined by the senior embryologists, as well as within day of biopsy categories according to the hpi. Finally, a receiver operating characteristic (ROC) curve analysis was conducted to assess the area under the curve (AUC) for the discrimination of euploidy based on the embryologists’ assessment and based on iDAScore v1.0. Overall, 610 patients conducted ≥1 vitrified-warmed euploid blastocyst SET by the time of paper drafting (N = 808 SETs). iDAScore v1.0 was finally assessed for its association with the outcome after euploid SET (i.e., either no LB or LB) (Figure 1). A sub-analysis of iDAScore v1.0 in these two groups was conducted within blastocyst quality categories as defined by the senior embryologists, as well as within day of biopsy categories according to the hpi. At last, a ROC curve analysis was conducted to assess the AUC for the discrimination of a LB after euploid SET based on embryologists’ assessment and based on iDAScore v1.0.

Beyond the association studies, we also evaluated what would have been the impact of iDAScore v1.0 had it been used clinically to prioritize the blastocyst for transfer, either without or with the diagnostic information derived from aneuploidy testing. The first simulation was conducted in the 587 cycles (N = 587/1232, 47.6% of the cycles included) where sibling ≥1 euploid and ≥1 aneuploid blastocysts were diagnosed. Specifically, we calculated how often the embryologists and iDAScore v1.0 would have blindly graded a euploid or aneuploid blastocyst as top quality within each cohort (Figure 2A). The second simulation was carried out in 202 cycles conducted up to December 2021 (N = 202/1057, 19% of the cycles included) that, before we drafted this manuscript, had already achieved ≥1 LB from a cohort of ≥2 euploid blastocysts. Specifically, we calculated: (i) how often the embryologists and iDAScore v1.0 would have been equally effective in prioritizing the euploid blastocyst to transfer that, indeed, resulted in a LB (that is, embryologists and iDAScore v1.0 would have been equally effective), (ii) how often iDAScore v1.0 would have selected a euploid blastocyst for ET that resulted in a LB, but the embryologists transferred this embryo only after another one that did not result in a LB (that is, iDAScore v1.0 would have involved an earlier LB), and (iii) how often iDAScore v1.0 would have selected a euploid blastocyst that did not result in a LB and that was transferred by the embryologists only after another embryo which instead did result in a LB (that is, iDAScore v1.0 would have involved a later LB) (Figure 2B). 

### 2.2. IVF Protocols

Only the first IVF cycles in EmbryoScope incubators and PGT-A were included. Ovarian stimulation was conducted only with GnRH antagonist protocols and ovulation was triggered with either GnRH-agonist or hCG [39,40,41]. Oocytes were retrieved 35 h after trigger, and only ICSI was conducted as previously detailed [42]. Only continuous embryo culture was conducted in a continuous single culture medium (CSCM, Irvine Scientific, USA) with a refresh on day 5 in case of extended culture to day 6–7. Laser-assisted TE biopsy was conducted according to the “simultaneous zona pellucida (ZP) opening and biopsy method” [13,43]. This approach does not entail any ZP drilling at the cleavage stage (i.e., day 3 of preimplantation development), and the embryos are left undisturbed until their full expansion on day 5–7 [14]. Only CCT analyses were conducted [44,45,46] to identify non-mosaic full chromosome aneuploidies, and chromosome intermediate copy numbers (ICN) were reported as either euploid or aneuploid based on a 50% threshold according to the assessment of our reference genetic laboratory (Igenomix, Italy) [47,48]. Indeed, the report of putative mosaicism based on ICN < 50% has been shown clinically ineffective in a recent blinded non-selection study [49]. Vitrification was conducted within 90 min from biopsy [43]. Only euploid blastocyst SETs were performed 2 h after warming in a following menstrual cycle. Endometrial preparation was managed with either a modified-natural cycle or through hormone replacement therapy [41]. All SETs from the same oocyte retrieval cycle were included. 

Blastocyst morphology was graded by senior embryologists based on the Gardner’s scoring system [4]. Specifically, the ICM was graded “A” in case of a structure characterized by several strictly packed cells, “B” in case of a discernible structure with several but roughly packed cells, or “C” in case of a structure difficult to distinguish with few low-quality cells. Similarly, the TE was graded “A” in case of a well-organized cohesive epithelium with several cells, “B” in case of a loose epithelium with few cells, or “C” in case of very few and/or low-quality cells. Each blastocyst was graded in real time by two senior embryologists (Fleiss’ Kappa for ICM morphology assessment = 0.610, i.e., good agreement; Fleiss’ Kappa for TE morphology assessment = 0.806, i.e., excellent agreement). In case of disagreement, a third senior embryologist decided the grade. Our internal grading scheme clusters all “AA” blastocysts within the “excellent” quality category, “AB” and “BA” blastocysts within the “good” quality category, “BB”, “AC”, and “CA” within the “average” quality category, and “CC”, “BC”, and “CB” within the “poor” quality category [13]. Whenever ≥2 blastocysts were obtained in a cohort, the senior embryologists would identify the top-quality embryo based on: (i) its overall quality, (ii) the time of biopsy in hpi (the earlier, the better), (iii) the TE quality, and (iv) the expansion (the larger, the better). 

### 2.3. iDAScore v1.0

The DL model iDAScore v1.0 is based on a 3D convolutional neural network [31,50]. The model was trained on a large data set from 18 clinics worldwide containing a total of 115,832 embryos. Of them, 14,644 embryos were transferred on day 5 or later, resulting in 4337 positive fetal heartbeats and 10,307 implantation failures. The input to the model is 128 images sampled one hour apart covering the time from 12 hpi to 140 hpi. No patient data (e.g., age) or morphokinetic parameters are used as input to the model. The model is intended to be used on all embryos without any pre-selection and as a decision support system where the final decision is made by the user. The software, which is an add-on to the existing EmbryoSuite software (Vitrolife A/S), generates for each embryo a score between 1.0 (lowest) and 9.9 (highest), which is meant to express its implantation potential. Clinica Valle Giulia (GeneraLife IVF) was not involved in training the model; therefore, this study should be considered an independent external pre-clinical validation. 

### 2.4. Statistical Analysis

Continuous variables were reported as mean ± SD and Shapiro–Wilk test was adopted to assess a Gaussian distribution of the data. Mann–Whitney U, Kruskal–Wallis, Student’s t-tests or ANOVA were adopted to define significant differences among each comparison. Fisher’s exact or chi-squared tests were instead adopted for categorical variables. Linear and logistic regressions were conducted to confirm significant associations. Putative confounders (relevant to patients, embryos, and cycle characteristics) were outlined through univariate analyses and eventually included to adjust the results in multivariate analyses. All statistical analyses were performed with the software SPSS (IBM, Armonk, NY, USA). Post-hoc statistical power analyses were conducted via G*Power for all the main comparisons.

## 3. Results

### 3.1. The Patients Included Are Predominantly Poor Prognosis and of Advanced Maternal Age 

The patients included in this study represent the average population of women undergoing IVF at our center, predominantly advanced maternal age (38.7 ± 3.4 years) and poor prognosis (2.9 ± 1.8 blastocysts biopsied, of which 1.2 ± 1.3 euploids) (Appendix A).

### 3.2. A Generally Good Association Exists between the Conventional Parameters of Morphological Evaluation and iDAScore v1.0

iDAScore v1.0 was significantly associated with the day of full blastocyst development (day 5 blastocysts, N = 1462, 8.2 ± 1.5 versus day 6 blastocysts, N = 1874, 5.6 ± 1.7 versus day 7 blastocysts, N = 268, 3.9 ± 1.4; *p* < 0.01 and power = 99% for all comparisons; Figure 3A). The same data are presented in Appendix A as a dispersion plot that associates the time of biopsy of each embryo with its iDAScore v1.0. The linear regression analysis confirmed the significant association (unstandardized coefficient B: −0.092, 95% CI from −0.096 to −0.089, *p* < 0.01). Nevertheless, although the mean and median values are significantly different across the groups, long tails were shown in both graphs around low iDAScore v1.0 values in the day 5 group, as much as around high iDAScore v1.0 in the day 7 group. Conversely, day 6 blastocysts show a more widespread distribution of the data.

iDAScore v1.0 was also significantly associated with ICM quality (A grade, N = 2107, 7.5 ± 1.8 versus B grade, N = 833, 5.6 ± 1.9 versus C grade, N = 664, 4.4 ± 1.7; *p* < 0.01 and power >99% for all comparisons; Figure 3B). iDAScore v1.0 was significantly associated with TE quality as well (A grade, N = 1988, 7.5 ± 1.8 versus B grade, N = 951, 5.9 ± 1.9 versus C grade, N = 664, 4.3 ± 1.6; *p* < 0.01 and power >99% for all comparisons; Figure 3C). Additionally, in this analysis, long tails were shown in data distribution according to A and C grades, while B grade ICM/TE were associated with a more widespread distribution.

Conventionally, according to our internal grading method [13], AA blastocysts are considered of excellent quality; AB and BA of good quality; BB, AC, and CA of average quality; and CC, BC, and CB of poor quality. iDAScore v1.0 mirrors this clustering, as shown in Figure 4, except for a slight propensity to weigh the TE as more relevant than the ICM. In fact, in the “average quality” cluster, CA blastocysts (N = 14, 6.3 ± 1.5) showed iDAScore v1.0 higher than BB blastocysts (N = 446, 5.6 ± 1.8, *p* = 0.02). In the “poor quality” (lower than BB) cluster, CC blastocysts (N = 483, 4.1 ± 1.5) resulted in an iDAScore v1.0 lower than both BC and CB ones (N = 162, 4.6 ± 1.6 and N = 167, 5.1 ± 2.0, respectively; *p* = 0.05 and *p* < 0.01, respectively; Figure 4). iDAScore v1.0 within each blastocyst morphology group also decreases according to the time of biopsy, with sharper decreases in the good and average quality groups and milder decreases in the excellent and poor quality ones (Appendix A).

It is interesting that iDAScore v1.0 also slightly decreases according to maternal age at oocyte retrieval (unstandardized coefficient B: −0.036, 95% CI from −0.057 to −0.015, *p* < 0.01; Appendix A). Although clinically irrelevant, as this tool is intended to prioritize the blastocyst for transfer within a cohort of siblings (i.e., deriving from equally aged oocytes), this correlation supports a general association between advanced maternal age, poorer blastocyst morphology, and lower competence. 

### 3.3. iDAScore v1.0 Is Significantly Associated with Euploidy, but the AUC Is 0.60

Euploid blastocysts showed significantly higher iDAScore v1.0 (N = 1443, 7.0 ± 2.1) than single (N = 1194, 6.5 ± 2.2, *p* < 0.01 and power > 99%) and especially complex aneuploid embryos (N = 967, 5.8 ± 2.1, *p* < 0.01 and power > 99%; Figure 5A). Indeed, a logistic regression analysis adjusted for maternal age confirmed an association (multivariate OR: 1.18, 95% CI 1.14–1.22, *p* < 0.01) (Table 1). In addition, a good association was shown with the conventional parameters of embryo grading as reported in Table 2. The ROC curve analysis, in fact, highlighted an AUC of 0.66 (95% CI 0.64–0.68) for the discrimination between embryologists’ assessment and euploidy, and a lower AUC of 0.60 (95% CI 0.59–0.62) for iDAScore v1.0 (Figure 5B). Of note, iDAScore v1.0 decreases rather uniformly according to the time of biopsy in the groups euploid, single, and complex aneuploid (Appendix A). There is no additional discrimination due to iDAScore v1.0 between euploid and aneuploid blastocysts within embryo quality groups as defined by the embryologists (Appendix A). Conversely, within the day of biopsy groups (5 and 6), significantly different iDAScore v1.0 were still observed between euploid and aneuploid embryos (Appendix A).

### 3.4. When Both Euploid and Aneuploid Embryos Were Diagnosed from the Same Cohort, iDAScore v1.0 Ranked the Euploid Blastocyst on Top in 63% of the Cases

In 587 cycles, both euploid and aneuploid blastocysts were diagnosed. According to the embryologists’ assessment, the blastocysts ranked as top quality in their cohort of siblings were euploid in 47% of the cases and aneuploid in 24% of the cases. In the remaining 29% of the cases, euploid and aneuploid blastocysts were equally ranked as top quality (Figure 6A). According to iDAScore v1.0, in 63% and 37% of the cases, respectively, a euploid and an aneuploid blastocyst would have been ranked as top quality (Figure 6B). In the latter simulation, it is indeed unlikely that two or more blastocysts would have the same score. In fact, a 0.1 difference is sufficient to rank a blastocyst as better than another. 

### 3.5. iDAScore v1.0 Is Significantly Associated with the Achievement of a LB after Euploid Blastocyst SET, with a 0.66 AUC

LB showed significantly higher iDAScore v1.0 (N = 361, 7.6 ± 1.8) than no LB (N = 447, 6.5 ± 2.2, *p* < 0.01 and power > 99%; Figure 7A), and logistic regression analysis confirmed this association (OR: 1.3, 95% CI 1.2–1.4, *p* < 0.01) (Table 1). Nevertheless, a good association was also shown with the conventional parameters of embryo grading and the day of biopsy, as described in Table 2. The ROC curve analyses, in fact, were almost superimposable: AUC 0.64 (95% CI 0.60–0.67) for the association between embryologists’ assessment and euploidy, and AUC 0.66 (95% CI 0.62–0.69) for iDAScore v1.0 (Figure 7B). Interestingly, a larger reduction in iDAScore v1.0 was reported according to the time of biopsy in the group “no LB” with respect to the group “LB” (Appendix A). In addition, significantly higher iDAScore v1.0 characterized the blastocysts resulting in a LB versus the ones that did not also for both the sub-analyses within: (i) embryo morphology groups as outlined by the embryologists, except for the group “<BB” (Appendix A), and (ii) day of biopsy groups as outlined by the hpi, except for the group “day7” (Appendix A).

### 3.6. When at Least Two Euploid Blastocysts Were Available from the Same Cohort, the Embryologists Would Have often Disagreed with iDAScore v1.0 on Their Ranking 

In 202 cycles, at least two euploid blastocysts were available for transfer (the raw data are shown in Appendix A). In 52% of these cases, the embryologists and iDAScore v1.0 would have been equally effective since they would have either agreed on the blastocyst to prioritize for transfer, and that resulted in a LB, or they would have disagreed, but both would have been correct (Appendix A). In 15% of the cases, iDAScore v1.0 would have identified the competent embryo earlier than the embryologists, while in 3% of the cases, iDAScore v1.0 would have identified the competent embryo later than the embryologists (Appendix A). Nevertheless, this simulation is partially biased, because in 29% of the cases iDAScore v1.0 putative influence could not be assessed. Specifically, in discordant cases where the embryologists’ choice for transfer resulted in a LB, but the highest ranked blastocysts according to iDAScore v1.0 had not been transferred, so their reproductive competence is unknown (Appendix A). Consequently, the rate of equal and poorer effectiveness instances of iDAScore v1.0 in relation to the embryologists might be higher.

## 4. Discussion

AI and automation will strongly impact the future of IVF, meeting the needs for standardization and lower workload in the laboratories [11,32,51,52]. Nonetheless, AI-powered tools for embryo selection purposes requires further refinement, as most studies in this field show significant and recurrent limitations: (i) the nature of the training datasets is not representative of all clinical practices, (ii) the use of clinical pregnancy or fetal heartbeat as an endpoint rather than LB, (iii) the low sample sizes, and (iv) the lack of multicenter validation data [30,37]. In this study, we aimed at validating iDAScore v1.0 in ICSI cycles with TE biopsy, CCT analysis and vitrified-warmed euploid blastocyst SETs. We assessed: (i) its association with embryo morphology, day of development, euploidy, and LB, and (ii) its putative clinical utility in a retrospective simulation.

As previously reported by other studies, iDAScore v1.0 demonstrated a good correlation with the morphological parameters assigned by experienced embryologists to each blastocyst, either for ICM *per se* and TE *per se*, or for overall blastocyst morphology [31,53]. Notably, whenever the ICM and the TE of the same blastocyst were reported of a different quality (e.g., AC versus CA, BC versus CB), iDAScore v1.0 favored the latter. This trend inherently advocates a better predictivity of TE quality upon embryo implantation, as already suggested previously from other groups not using AI-powered tools [54,55,56,57,58,59]. Moreover, embryos achieving full blastocyst expansion on day 5 (<120 hpi) showed higher iDAScore v1.0 than embryos reaching that same stage on day 6 (121–144 hpi) or 7 (>144 hpi), also suggesting a better quality for the former, consistent with our previous analysis based on a different AI tool [14]. These data overall support the use of iDAScore v1.0 to objectify blastocyst morphological assessment within and between IVF clinics, as well as between professionals working at different laboratories, regardless of their experience, social, economic, clinical, and regulatory contexts, potentially influencing clinical choices [16,17,60]. Of note, the performance of iDAScore v1.0 in the evaluation of day 7 and/or blastocysts lower than BB might be suboptimal. In fact, the last frame analyzed by the software in its current version is at 140 hpi, thus it will not capture embryo development on day 7. In addition, poor-quality blastocysts are often deselected by embryologists worldwide, and thus are not sufficiently represented in the training data set. Future versions of iDAScore may benefit from datasets enriched in these populations of embryos and, so far, an early validation of v2.0 already shows significantly improved model performance in comparison to v1.0 with extended image analysis up to 148 hpi [61].

Although a large proportion of excellent quality blastocysts (AA) in a general population of advanced maternal age women are aneuploid (≈50%), while ≈25% of poor quality ones (lower than BB) might be euploid [15], a significant association exists between embryos’ morphology and their chromosomal and reproductive competence [6,7,8]. Therefore, we tested iDAScore v1.0 in our dataset for its association with euploidy, to investigate whether this software may play a role in prioritizing euploid blastocysts for ET. Indeed, invasive PGT-A is still the only approach to reliably deselect blastocysts diagnosed with full chromosome aneuploidies and achieve a negative predictive value as high as 98% (i.e., lethality rate when aneuploid ETs were conducted in blinded non-selection studies) [62]. Yet, PGT-A is not universally applicable due to regulations, costs, and expertise; therefore, the long-lasting quest for non-invasive biomarkers of euploidy has also recently focused on AI-powered morphological and morphodynamic assessments [8,63,64,65,66]. Here, we report a significant association between iDAScore v1.0 and euploidy, even after results are clustered according to the day of full blastocyst expansion (day 5–7). The same result is not achieved for excellent, good, average, and poor blastocyst morphology clusters, as defined by conventional embryologists’ assessment. This sub-analysis explains why the AUC for euploidy prediction derived from embryologist’s evaluation performs better than that of iDAScore v1.0. It must be said, though, that the former approach is limited by its intrinsic subjectivity and limited generalizability [11] as well as by its poor ranking potential (i.e., based on three ICM morphology classes and three TE morphology classes in day 5, 6 or 7 after insemination). Conversely, the latter is more objective and reproducible, and leverages on a score difference as low as 0.1 to discriminate between embryos of slightly different quality. Indeed, among those cycles with at least one euploid and one aneuploid sibling blastocysts (about 50% of the cycles conducted during the study period), iDAScore v1.0 would have ranked euploid embryos as top quality in 63% of the cases versus 47% of the cases for embryologists’ assessment. This latter estimate was indeed influenced by 29% of cases where euploid and aneuploid blastocysts were both tied as top quality according to embryologists’ assessment.

Of note, iDAScore v1.0 was trained to predict implantation, not euploidy, and although (except for vital aneuploidies) a blastocyst must be euploid to result in a LB, as many as 50% of euploid blastocysts typically fail to implant [20,21,22]. It is therefore remarkable that iDAScore v1.0 showed a more evident association with LB in the context of euploid blastocyst SETs than with euploidy *per se*, consistent with recent evidence shown by a Japanese group for untested embryos [67]. It is promising that the AUC for the LB outcome mirrored the AUC resulting from embryologists’ assessment. It is also reassuring that significantly higher iDAScore v1.0 output was observed for euploid blastocysts resulting in a LB versus reproductively incompetent ones among embryo quality and day of development clusters. These observations may suggest an additional application for iDAScore v1.0, namely it may be the swing vote among sibling euploid blastocysts to rank the embryos for ET. To this end, we conducted a second simulation in all the cycles where at least two euploid blastocysts and at least one LB were obtained (about 20% of the cycles included). This was aimed at understanding how often iDAScore v1.0 would have modified embryologists’ choice, had it been used clinically. Notably, a different choice would have occurred in about 50% of the cases, suggesting a concrete influence of this tool, also in the context of PGT-A cycles. According to our data, iDAScore v1.0 would have involved delayed LB outcome in 3% of the cases, and an earlier one in 15% of the cases. However, this simulation may be unbalanced in favor of the software over the operators, because in 59 cases (29%) iDAScore v1.0 putative influence could not be assessed. Specifically, in these cycles the embryologists transferred euploid blastocysts which resulted in a LB, but which were not graded as top quality by iDAScore v1.0, while top scoring blastocysts were not yet transferred at the time of drafting of this paper. On the contrary, the cases where iDAScore v1.0 would have outperformed the embryologists could all be computed, as the incorrect choice of the latter (i.e., no LB achieved) was always evident. Nevertheless, we chose to report these preliminary data here because they represent relevant (although only observational) evidence of the potential contribution of this tool for embryo selection purposes beyond euploidy. An RCT comparing embryologists with iDAScore performance per ET is certainly needed now to assess the true clinical contribution of this tool for embryo selection purposes.

## 5. Conclusions

Several embryo evaluation tools based on AI technologies have been proposed in IVF to date. For instance, the Spatial–Temporal Ensemble Model (STEM) and its upgrade, STEM+, promisingly reported to be able to predict blastocyst formation with high accuracy and AUC [68]. In a previous work from our group, we reported good consistency between an AI-powered software named CHLOE^TM^ and blastocyst quality as defined by clinical embryologists [14]. IVY, a deep learning model producing a score between 0 and 1, was also tested for its prediction of the likelihood of blastocyst implantation and showed encouraging results [50]. Lately, then, some tools have been assessed for a putative prediction of euploidy. Embryo Ranking Intelligent Classification Algorithm (ERICA), for instance, outperformed clinical embryologists in ranking euploid blastocysts as top quality in their cohorts [33], and others such as Euploid Prediction Algorithm (EPA), STORK-A and iDAScore v1.0 itself showed good correlation with euploidy [65,66,69]. Nonetheless, in our view, present and future AI-powered tools should be aimed at supporting embryologists in prioritizing for (either untested or euploid) transfer the embryo(s) more likely to result in a live birth in their cohort, rather than at predicting euploidy (e.g., [70]). The accurate diagnosis of euploidy, as of today, still requires comprehensive chromosome testing technologies—with no report of mosaicism based on ICN—and invasive TE biopsy sampling approaches. Most importantly, although it is essential, euploidy is not sufficient to obtain a healthy baby, and the prediction of this latter outcome—and not of euploidy—should be the main aim of any embryo selection tool. This manuscript summarizes an independent, external, pre-clinical validation of iDAScore v1.0, one of the currently available AI-powered software programs for embryo selection. Within the limitations of retrospective design, our data support iDAScore as a promising tool to objectify embryo evaluation across embryologists and clinics, while preventing time-consuming and potentially biased morphokinetic manual annotations. The current v1.0 model performance upon euploidy and LB after euploid SETs is equivalent to embryologists’ performance. Nonetheless, this can be considered a positive result for at least two reasons: (i) iDAScore was not trained to address these outcomes, and (ii) the AUC would be independent from embryologists’ subjective assessment and increase objectivity and reproducibility. We also provided preliminary evidence of the current clinical utility of iDAScore v1.0, had it been used for embryo ranking purposes. We advocate for a prospective, possibly multicenter, study to confirm our data with an RCT design. A cost-effectiveness analysis is desirable as well, which should include information about lab workload with and without this tool.

## Figures and Tables

**Figure 1 jcm-12-01806-f001:**
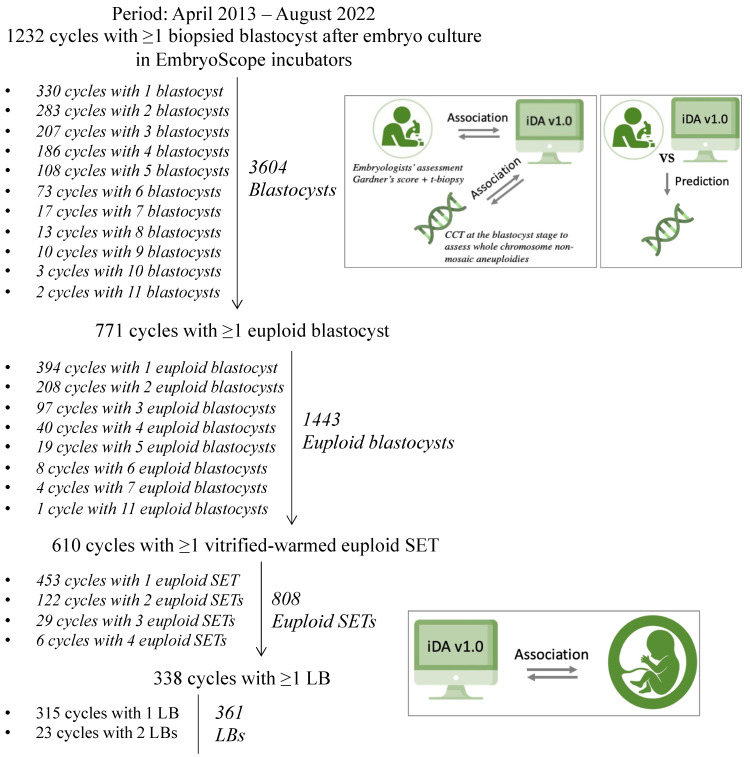
Association and prediction study workflow. T-biopsy, time of biopsy; CCT, comprehensive chromosome testing; iDA v1.0, Intelligent Data Analysis score version 1.0; SETs, single embryo transfers; LBs, live births.

**Figure 2 jcm-12-01806-f002:**
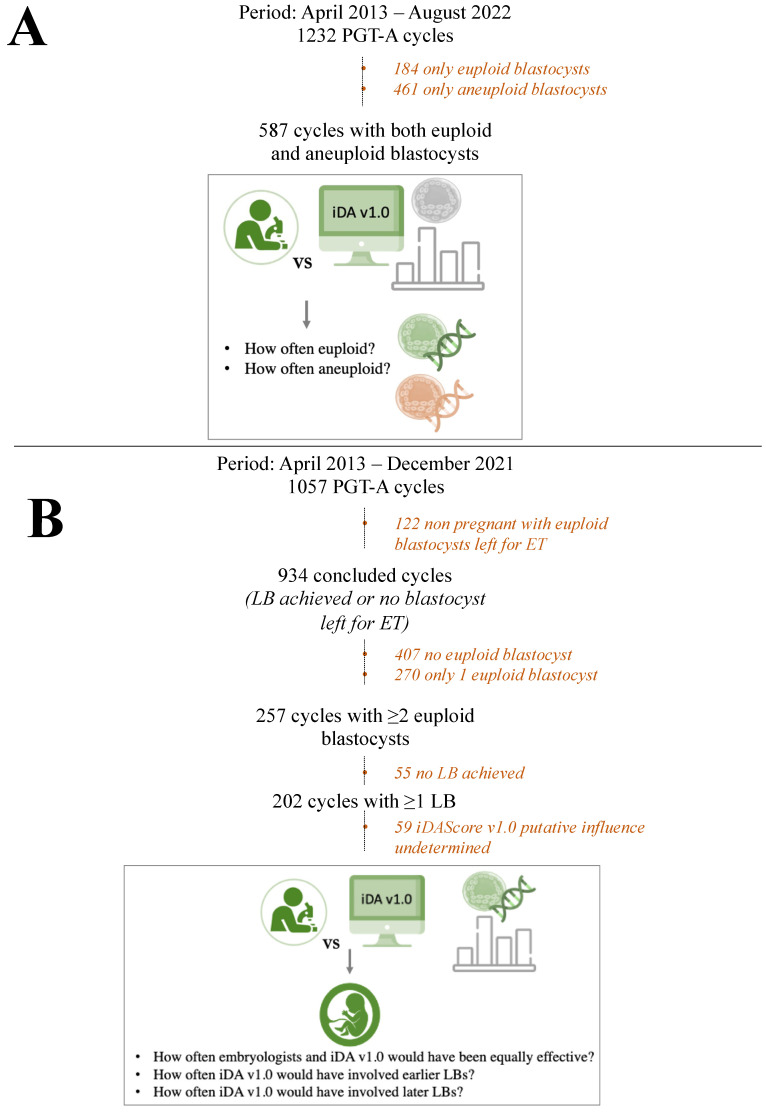
Clinical utility study workflow. (**A**) Definition of top-quality blastocysts within each cohort according to the embryologists versus iDAScore v1.0: how often were they euploid and how often aneuploid? (**B**) Definition of top-quality euploid blastocysts within each cohort according to the embryologists versus iDAScore v1.0: how often would they have been equally effective? How often would iDAScore v1.0 have involved an earlier live birth (LB)? How often would iDAScore v1.0 involved a later LB? In both figures, the orange phrases summarize the excluded cycles with the reasons for exclusion. PGT-A, preimplantation genetic testing for aneuploidies; ET, embryo transfer.

**Figure 3 jcm-12-01806-f003:**
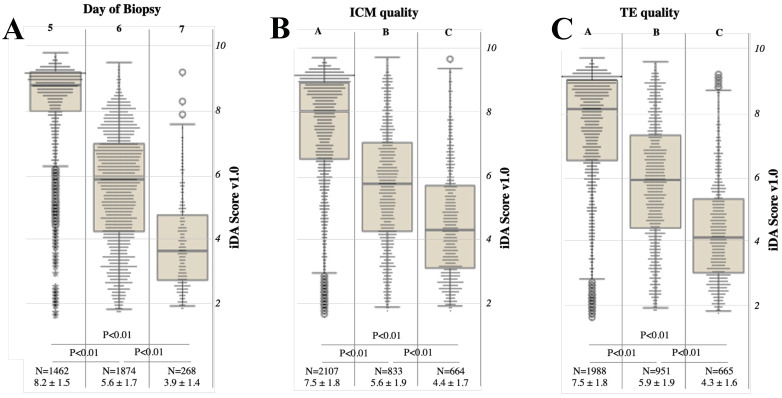
iDAScore v1.0 is associated with the day of biopsy (**A**), the inner cell mass (ICM) quality (**B**), and the trophectoderm (TE) quality (**C**). The day of biopsy is defined according to hours between insemination and achievement of a grade of blastocyst expansion compatible with a TE biopsy: ≤120 h post insemination (hpi) = day 5, 121–144 hpi = day 6, >144 hpi = day 7. ICM and TE quality were defined according to Gardner’s score as A, B, or C (from best to worst quality).

**Figure 4 jcm-12-01806-f004:**
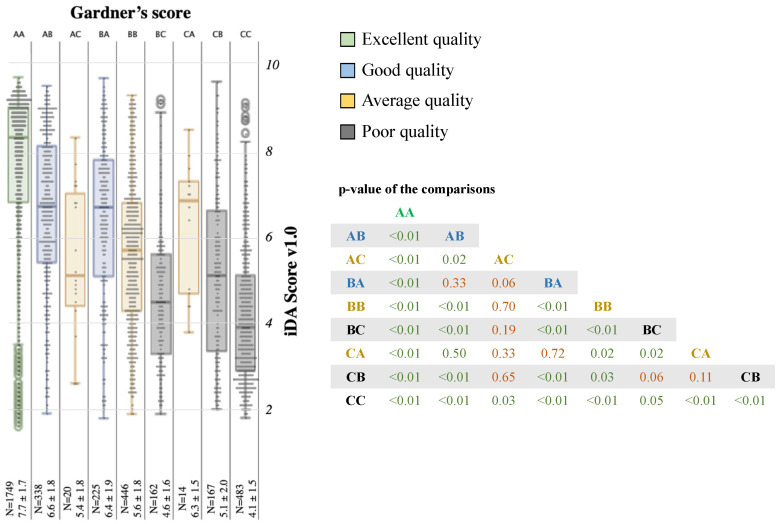
iDAScore v1.0 is associated with overall blastocyst quality. A, B, and C (from best to worst quality) outline inner cell mass (the first letter) and trophectoderm (the second letter) quality as defined by the embryologists according to the Gardner’s score. Overall blastocyst quality is defined as excellent (AA, green), good (AB and BA, blue), average (AC, CA, and BB, gold), or poor (BC, CB, and CC, grey) according to Gardner’s score adapted by Capalbo et al. [13]. The table summarizes the *p*-values of each sub-group comparison.

**Figure 5 jcm-12-01806-f005:**
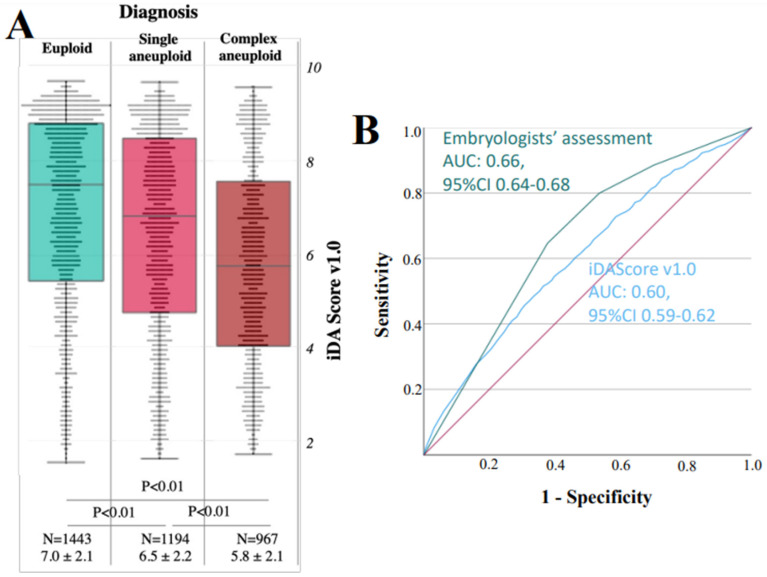
iDAScore v1.0 is associated with blastocysts’ chromosomal constitution, but the AUC is 0.60. (**A**) Association between iDAScore v1.0 and blastocysts’ chromosomal constitution clustered as euploid, single aneuploid, or complex aneuploid; (**B**) Receiver operating characteristic (ROC) curve analysis. The green curve represents the discrimination of embryologists’ assessment upon euploidy with an area under the curve (AUC) of 0.66, while the blue curve represents the discrimination of iDAScore v1.0 upon euploidy with an AUC of 0.60.

**Figure 6 jcm-12-01806-f006:**
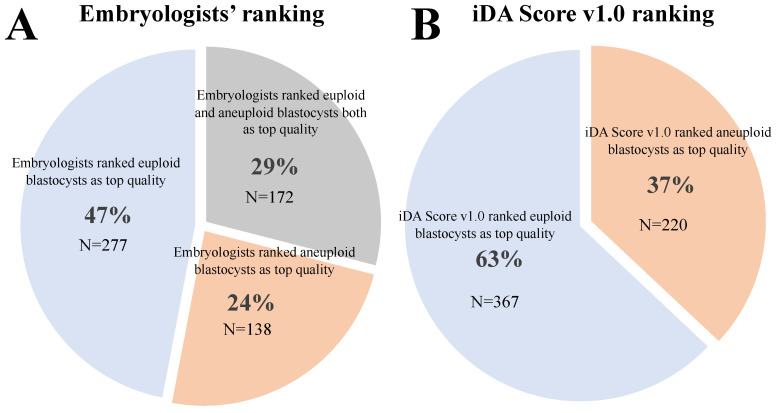
Association between the top-ranked blastocysts within each cohort according to: (**A**) the embryologists’ ranking, and (**B**) iDAScore v1.0 ranking and their chromosomal constitution.

**Figure 7 jcm-12-01806-f007:**
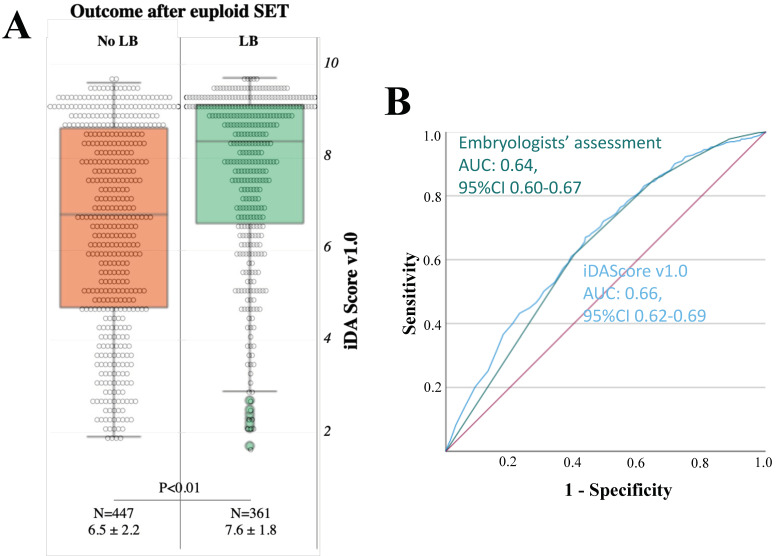
iDAScore v1.0 is associated with live birth (LB) after euploid blastocysts single embryo transfer (SET), but the AUC is 0.66. (**A**) Association between iDAScore v1.0 and a negative (no LB) or positive (LB) clinical outcome; (**B**) Receiver operating characteristic (ROC) curve analysis. The green curve represents the discrimination of embryologists’ assessment upon a LB after euploid blastocyst SETs with an area under the curve (AUC) of 0.64, while the blue curve represents the discrimination of iDAScore v1.0 upon a LB after euploid blastocyst SETs with an AUC of 0.66.

**Table 1 jcm-12-01806-t001:** Logistic regressions for the association between iDAScore v1.0 with euploidy (adjusted for maternal age) and live birth (LB) after euploid single embryo transfer (SET).

**Outcome: euploidy**	**Univariate** **OR, 95% CI, *p*-value**	**Multivariate-OR, 95% CI, *p*-value**
Maternal age	0.82, 95% CI 0.8–0.84, *p* < 0.01	0.82, 95% CI 0.8–0.84, *p* < 0.01
iDAScore v1.0	1.18, 95% CI 1.14–1.22, *p* < 0.01	1.18, 95% CI 1.14–1.22, *p* < 0.01
**Outcome: LB per euploid SET**	**Univariate OR, 95% CI, *p*-value**	**-**
iDAScore v1.0	1.30, 95% CI 1.2–1.4, *p* < 0.01	-

**Table 2 jcm-12-01806-t002:** Logistic regressions for the association between embryologists’ assessment with euploidy (adjusted for maternal age) and live birth (LB) after euploid single embryo transfer (SET).

**Outcome: euploidy**	**Univariate OR, 95% CI, *p*-value**	**Multivariate OR, 95% CI, *p*-value**
Maternal age	0.82, 95% CI 0.8–0.84, *p* < 0.01	0.82, 95% CI 0.8–0.84, *p* < 0.01
Blastocyst quality:		
AA	-	-
AB, BA	0.57, 95% CI 0.47–0.69, *p* < 0.01	0.57, 95% CI 0.47–0.71, *p* < 0.01
BB, AC, CA	0.30, 95% CI 0.24–0.38, *p* < 0.01	0.32, 95% CI 0.25–0.40, *p* < 0.01
CC, BC, CA	0.23, 95% CI 0.19–0.27, *p* < 0.01	0.25, 95% CI 0.2–0.31, *p* < 0.01
Day of biopsy:		
5	-	-
6	0.62, 95% CI 0.54–0.72, *p* < 0.01	1.02, 95% CI 0.87–1.2, *p* = 0.81
7	0.34, 95% CI 0.25–0.45, *p* < 0.01	0.78, 95% CI 0.55–1.1, *p* = 0.16
**Outcome: LB per euploid SET**	**Univariate OR, 95% CI, *p*-value**	**Multivariate OR, 95% CI, *p*-value**
Blastocyst quality:		
AA	-	-
AB, BA	0.61, 95% CI 0.40–0.94, *p* = 0.02	0.72, 95% CI 0.46–1.11, *p* = 0.14
BB, AC, CA	0.39, 95% CI 0.22–0.70, *p* < 0.01	0.50, 95% CI 0.28–0.90, *p* = 0.02
CC, BC, CA	0.18, 95% CI 0.09–0.35, *p* < 0.01	0.24, 95% CI 0.12–0.47, *p* < 0.01
Day of biopsy:		
5	-	-
6	0.48, 95% CI 0.36–0.64, *p* < 0.01	0.59, 95% CI 0.44–0.81, *p* < 0.01
7	0.26, 95% CI 0.11–0.63, *p* < 0.01	0.47, 95% CI 0.19–1.18, *p* = 0.11

## Data Availability

Data are contained within the article and Appendix A.

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
