# Peer review of "Towards Automation in IVF: Pre-Clinical Validation of a Deep Learning-Based Embryo Grading System during PGT-A Cycles"

_jcm, 2023, doi:10.3390/jcm12051806_

Round 1

Reviewer 1 Report

The manuscript entitled: “Towards automation and standardization in IVF: pre-clinical validation of a deep learning-based embryo grading system during PGT-A cycles” presents a retrospective study regarding the clinical utility of an AI-based prediction model, the iDAscore v1.0. The study provides significant contribution to the field, employing a sound methodology. However, several points should be clarified prior to this study being accepted for publication.

The introduction is well-written and provides an adequate background. In the materials and methods section, the associations between embryologists and AI-based embryo grading system are sufficiently described, providing figures with the study workflow. Despite the large number of figures included, the authors do not mention the method employed for the sample size estimation (i.e. power analysis).

In the results section, the authors conduct several correlations between embryologist’s assessment and iDAscore v1.0. It may be appropriate that more data regarding the predictive value of the current AI prediction model, like sensitivity, specificity, PPV and NPV be provided. Especially for the outcome of euploidy, since the scientific community is moving towards a less invasive approach compared to the current gold standard, these outcomes will be significantly informative for the readership. Additionally, in the results section the text could be separated in distinct subunits, based on the different associations performed. This format will assist the readership who may not be familiar with advanced statistical methods. As a minor comment the (±) symbol is not shown in the first paragraph of the results section. It may be necessary to employ a different encoding.

In the discussion section the authors describe their findings and discuss them to current literature. However, a thorough comparison with other models that have been developed is not presented. The authors thoroughly discuss the limitations of their study. In the conclusions section, the authors mention that “v1.0 model performance… is equivalent to embryologists.” Nevertheless, it is not clear whether iDAscore v1.0 constitutes a cost-effective approach which could be included in IVF laboratory, enhancing clinical or embryological outcomes.

Author Response

Authors: We want to sincerely thank the editor and reviewers for their time and helpful comments. We revised the manuscript accordingly and hope it fulfils your expectations in its current form.

R1

The manuscript entitled: “Towards automation and standardization in IVF: pre-clinical validation of a deep learning-based embryo grading system during PGT-A cycles” presents a retrospective study regarding the clinical utility of an AI-based prediction model, the iDAscore v1.0. The study provides significant contribution to the field, employing a sound methodology. However, several points should be clarified prior to this study being accepted for publication. 

The introduction is well-written and provides an adequate background. In the materials and methods section, the associations between embryologists and AI-based embryo grading system are sufficiently described, providing figures with the study workflow. Despite the large number of figures included, the authors do not mention the method employed for the sample size estimation (i.e. power analysis). 

Authors: Thanks for this important request. We have added the post-hoc statistical power calculated for the main comparisons, and it was always >99%. We also revised the MM section accordingly.

In the results section, the authors conduct several correlations between embryologist’s assessment and iDAscore v1.0. It may be appropriate that more data regarding the predictive value of the current AI prediction model, like sensitivity, specificity, PPV and NPV be provided. Especially for the outcome of euploidy, since the scientific community is moving towards a less invasive approach compared to the current gold standard, these outcomes will be significantly informative for the readership.

Authors: Thanks for your comment. The aim of this study was to outline putative associations between iDAScore v1.0 as a continuous variable and the main IVF outcomes that define the competence of an embryo, i.e., euploidy and live birth. In addition, we added two simulations to outline whether iDAScore v1.0 would have changed embryologists’ assessment (and with what consequences), had it been used clinically to prioritize embryos for transfer among cohort where a choice was feasible (i.e., >1 [euploid] blastocyst available). Here we did not categorize the scores within clusters, or according to a given cut-off, aiming at assessing iDAScore v1.0 as a tool to diagnose aneuploid embryos or reproductively competent embryos. In other terms, the clinical utility of each score is evaluated with respect to ranking the sibling blastocysts within a cohort and not to evaluate the actual likelihood of euploidy or live birth. We respectfully think that reporting sensitivity, specificity, PPV and NPV would off-topic here and possibly misleading for the readers.

Additionally, in the results section the text could be separated in distinct subunits, based on the different associations performed. This format will assist the readership who may not be familiar with advanced statistical methods.

Authors: We agree with this reviewer. In fact, the results section is already structured in paragraphs with sub-headings (“The patients included are predominantly poor prognosis and of advanced maternal age”, “A generally good association exists between the conventional parameters of morphological evaluation and iDAScore v1.0”, “iDAScore v1.0 is significantly associated with euploidy, but the AUC is 0.60”, “When both euploid and aneuploid embryos were diagnosed from the same cohort”, “iDAScore v1.0 ranked the euploid blastocyst on top in 63% of the cases”, “iDAScore v1.0 is significantly associated with the achievement of a LB after euploid blastocyst SET, with a 0.66 AUC”, “When at least two euploid blastocysts were available from the same cohort, the embryologists would have often disagreed with iDAScore v1.0 on their ranking”)

As a minor comment the (±) symbol is not shown in the first paragraph of the results section. It may be necessary to employ a different encoding.

Authors: Thanks.

In the discussion section the authors describe their findings and discuss them to current literature. However, a thorough comparison with other models that have been developed is not presented.

Authors: We added the following paragraph to the conclusions: “Several embryo evaluation tools based on AI technologies have been proposed in IVF up to date. For instance, the Spatial–Temporal Ensemble Model (STEM) and its upgrade STEM+, promisingly reported predictive of blastocyst formation with high accuracy and AUC [69]. In a previous work from our group, we reported good consistency between an AI-powered software named CHLOETM and blastocyst quality as defined by clinical embryologists [14]. IVY, a deep learning model producing a score between 0 and 1, was also tested for its prediction of the likelihood of blastocyst implantation and showed encouraging results [50]. Lately, then, some tools were assessed for a putative prediction of euploidy. Embryo Ranking Intelligent Classification Algorithm (ERICA), for instance, outperformed clinical embryologists in ranking euploid blastocysts as top-quality in their cohorts [33], and others like Euploid Prediction Algorithm (EPA), STORK-A and iDAScore v1.0 itself showed good correlation with euploidy [66,67,70]. Nonetheless, in our view, present and future AI-powered tools should be aimed at supporting embryologists in prioritizing for (either untested or euploid) transfer the embryo(s) more likely to result in a live birth in their cohort, rather than at predicting euploidy (e.g., [71]). The accurate diagnosis of euploidy, as of today, still requires comprehensive chromosome testing technologies - with no report of ICN as mosaic - and invasive TE biopsy sampling approaches. Most importantly, although being essential, euploidy is not sufficient to obtain a healthy baby, and the prediction of this latter outcome - and not of euploidy – should be the main aim of any embryo selection tool.”

The authors thoroughly discuss the limitations of their study. In the conclusions section, the authors mention that “v1.0 model performance… is equivalent to embryologists.”Nevertheless, it is not clear whether iDAscore v1.0 constitutes a cost-effective approach which could be included in IVF laboratory, enhancing clinical or embryological outcomes.

Authors: Revised as follows “We advocate for a prospective, possibly multicenter, study to confirm our data with a RCT design. A cost-effectiveness analysis is desirable as well, which should include information about lab workload with and without this tool.”

R2

In the manuscript jcm-2147517“Towards automation and standardization in IVF: pre-clinical validation of a deep learning-based embryo grading system during PGT-A cycles” the authors investigate the concordance of the iDAScore embryo assessment with the grading by an embryologist and assessment of the euploidy prediction by the iDAScore. The manuscript is well-written and easy to read.  The manuscript will be of interest to the users of the time-lapse incubator users and specialists in the area of ART.

Comments to the authors:

  1. In the title of the article and in the abstract, you state that iDAScore is the tool of standardization.  However, it is not described nor thoroughly discussed in the manuscript.

Authors: Thanks for this very important comment. By “standardization” we meant “make objective, measurable”. Therefore, we removed the word “standardization” from the title and replaced the word “standardize” with “objectify” in the manuscript.

  1. As far as I know, the Embryoscope grading software is adaptable to the embryological assessment Thus, it is   What was the level of confidence in the setting of the time-lapse scoring in the lab? Was there a machine learning phase for the time-lapse scoring?

Authors: It is true that the EmbryoScope grading software (i.e. EmbryoViewer) optionally includes a feature for automatic estimation of morphokinetics like division timings and blastocyst gradings. The feature is called “Guided Annotation” and is a paid add-on to the EmbryoViewer software. However, our clinic did not buy this add-on and as such there is no machine-learning involved in annotation of morphokinetics. As written in the manuscript annotations were made solely by the embryologists. Moreover, the study did not include any timing annotation or assessment of abnormal cleavage patterns. We only outlined the association between iDAScore v1.0 at the blastocyst stage and the IVF outcomes indicative of embryo competence.

  1. Please, indicate how many senior embryologists performed the embryo grading.

Authors: Thanks for the question. We have now added this paragraph to the MM section “Each blastocyst was graded in real-time by two senior embryologists (Fleiss’ Kappa for ICM morphology assessment = 0.610, i.e., good agreement; Fleiss’ Kappa for TE morphology assessment = 0.806, i.e., excellent agreement). In case of disagreement, a third senior embryologist decided the grade.”

  1. Please, specify in embryologists had access to the time-lapse score prior to embryo assessment by its morphology as it might had influenced the embryo grading.

Authors: Thanks for this key comment. We added this sentence to both the abstract and MM sections “All videos were retrospectively assessed through the iDAScore v1.0 software to grade each blastocyst without influencing embryologists’ clinical evaluations and decision-making process.”

  1. There is a contradiction. L311 you indicate that “According to the embryologists’ ranking, the top-quality blastocyst in each cohort was euploid in 47% of the cases and aneuploid in 24% of the cases.” and in L 395 “Although a large proportion of excellent quality blastocysts (AA) are aneuploid ( 50%)”

Authors: The first sentence is saying that among all cohorts of blastocysts with at least two blastocysts with different diagnoses, the embryologists graded as top-quality an embryo that turned out to be euploid in 47% of the cases; conversely, the second sentence is saying that (in a general population of advanced maternal age women) about half of excellent quality blastocysts are euploid. We revised these sentences as follows “In 587 cycles both euploid and aneuploid blastocysts were diagnosed. According to the embryologists’ assessment, the blastocysts ranked as top-quality in their cohort of siblings were euploid in 47% of the cases and aneuploid in 24% of the cases.” And “Although a large proportion of excellent quality blastocysts (AA) in a general population of advanced maternal age women are aneuploid (»50%), while »25% of poor-quality ones (<BB) might be euploid”. Thanks also for this comment.

  1. L 414 Please, clarify, what is “day of full expansion classes”

Authors: Thanks. Changed as follows “based on 3 ICM morphology classes and 3 TE morphology classes in day5, 6 or 7 after insemination”

Reviewer 2 Report

In the manuscript jcm-2147517“Towards automation and standardization in IVF: pre-clinical 2 validation of a deep learning-based embryo grading system 3 during PGT-A cycles” the authors investigate the concordance of the iDAScore embryo assessment with the grading by an embryologist and assessment of the euploidy prediction by the iDAScore. The manuscript is well-written and easy to read.  The manuscript will be of interest to the users of the time-lapse incubator users and specialists in the area of ART.

Comments to the authors:

1.   In the title of the article and in the abstract, you state that iDAScore is the tool of standardization.  However, it is not described nor thoroughly discussed in the manuscript.

2.   As far as I know, the Embryoscope grading software is adaptable to the embryological assessment Thus, it is   What was the level of confidence in the setting of the time-lapse scoring in the lab? Was there a machine learning phase for the time-lapse scoring?

3.   Please, indicate how many senior embryologists performed the embryo grading.

4.   Please, specify in embryologists had access to the time-lapse score prior to embryo assessment by its morphology as it might had influenced the embryo grading.

5.   There is a contradiction. L311 you indicate that “According to the embryologists’ ranking, the top-quality blastocyst in each cohort was euploid in 47% of the cases and aneuploid in 24% of the cases.” and in L 395 “Although a large proportion of excellent quality blastocysts (AA) are aneuploid  ( 50%)”

6.   L 414 Please, clarify, what is “3 day of full expansion classes”

Author Response

(The authors gave the same response as above.)
